# Cross-Feeding of a Toxic Metabolite in a Synthetic Lignocellulose-Degrading Microbial Community

**DOI:** 10.3390/microorganisms9020321

**Published:** 2021-02-04

**Authors:** Jessica A. Lee, Alyssa C. Baugh, Nicholas J. Shevalier, Brandi Strand, Sergey Stolyar, Christopher J. Marx

**Affiliations:** 1NASA Ames Research Center, Moffett Field, CA 94035, USA; 2Department of Biological Sciences, University of Idaho, Moscow, ID 83844, USA; AlyssaBaugh@uga.edu (A.C.B.); nshevalier@yahoo.com (N.J.S.); bstrand18@gmail.com (B.S.); sstolyar@uidaho.edu (S.S.); 3Institute for Bioinformatics and Evolutionary Studies, University of Idaho, Moscow, ID 83844, USA; 4Institute for Modeling Collaboration and Innovation, University of Idaho, Moscow, ID 83844, USA; 5Department of Microbiology, University of Georgia, Athens, GA 30602, USA

**Keywords:** synthetic ecology, lignocellulose, formaldehyde, microbial communities, methylotrophy, *Methylorubrum extorquens*, *Pseudomonas putida*, *Cellulomonas fimi*, *Yarrowia lipolytica*

## Abstract

The recalcitrance of complex organic polymers such as lignocellulose is one of the major obstacles to sustainable energy production from plant biomass, and the generation of toxic intermediates can negatively impact the efficiency of microbial lignocellulose degradation. Here, we describe the development of a model microbial consortium for studying lignocellulose degradation, with the specific goal of mitigating the production of the toxin formaldehyde during the breakdown of methoxylated aromatic compounds. Included are *Pseudomonas putida*, a lignin degrader; *Cellulomonas fimi*, a cellulose degrader; and sometimes *Yarrowia lipolytica*, an oleaginous yeast. Unique to our system is the inclusion of *Methylorubrum extorquens*, a methylotroph capable of using formaldehyde for growth. We developed a defined minimal “Model Lignocellulose” growth medium for reproducible coculture experiments. We demonstrated that the formaldehyde produced by *P. putida* growing on vanillic acid can exceed the minimum inhibitory concentration for *C. fimi*, and, furthermore, that the presence of *M. extorquens* lowers those concentrations. We also uncovered unexpected ecological dynamics, including resource competition, and interspecies differences in growth requirements and toxin sensitivities. Finally, we introduced the possibility for a mutualistic interaction between *C. fimi* and *M. extorquens* through metabolite exchange. This study lays the foundation to enable future work incorporating metabolomic analysis and modeling, genetic engineering, and laboratory evolution, on a model system that is appropriate both for fundamental eco-evolutionary studies and for the optimization of efficiency and yield in microbially-mediated biomass transformation.

## 1. Introduction

The global economy relies substantially on fossil fuels as a source of carbon compounds with applications ranging from energy to medicine. The pressing need to reduce dependency on these nonrenewable sources has inspired interest in the development of sustainable energy and bioproduct feedstocks. Due to its availability and energy density, lignocellulosic biomass has long been a target for the bioproduction of fuels, bioplastics, and other commodity chemicals [1,2,3]. However, while biological methods for upcycling the cellulosic portion are well developed, a significant challenge to the economic feasibility of using lignocellulose for bioenergy is the chemical complexity of lignin and its recalcitrance to breakdown. 

Lignin can compose 15–40% of unprocessed plant matter; initial processing of lignin yields complex mixtures of aromatic compounds, which may vary between types of feedstock [4,5]. There is no known single organism capable of catabolizing every compound in this complex mixture, and degradation can result in toxic intermediates that inhibit the growth of some of the very organisms involved. For instance, lignin-derived aromatic compounds are heavily substituted by methoxy (-OCH_3_) groups, which are transformed to formaldehyde in the process of microbial degradation [6,7,8,9,10,11]. Chemical pretreatment of lignin requires extreme chemicals or heat and may result in the destruction of important organisms or enzymes. Lignin is, therefore, considered an economically ineffective waste product of the food and other commercial industries. For lignocellulose to compete with nonrenewable carbon feedstocks, efficient, cost-effective processes of transformation must be developed that are robust over time and adaptable to diverse feedstocks.

In nature, lignocellulose is consumed by complex and dynamic communities of microbes, where distinct catabolic niches allow symbiosis by crossfeeding and detoxification of dangerous intermediates. While industries may mimic this “natural” strategy by using complex, undefined microbial communities (as is done in wastewater treatment [12]), there are long-standing problems with an approach in which the chemical transformations are not fully understood, particularly when it comes to the efficient production of specific desired bioproducts.

To the same ends, with very different means, synthetic and systems biology research frequently attempts to build a single metabolic powerhouse: one well-understood, often genetically engineered species capable of carrying out the entire process (e.g., [13]). This strategy has the advantage of not requiring the careful balance of growth conditions tailored to multiple species, and it is theoretically possible to maintain a consistent community in a fermentation system. However, when the specific enzymes responsible for particular transformations—as in the case of many lignocellulose components—are unknown, or function poorly in organisms with well-developed systems, complex and multistep processes can be a challenge.

Incorporating the best of both systems, the synthetic ecology approach describes a highly defined and specifically engineered community of organisms [14,15,16,17]. By narrowing the number of organisms, compared to “natural” undefined communities, it is possible to create conditions where each can thrive, especially if organisms are engineered or evolved for their roles. It is also possible, in contrast to the “powerhouse” strain approach, to select organisms with native affinities and tolerances suited for their role that are complex to engineer, including resistances to heat, acid, toxic intermediates, or the ability to store carbon or biomass in a form that can be used economically in downstream processes.

In this work, we describe a synthetic microbial community designed for the degradation of lignocellulose, with a particular focus on addressing the problem of a toxic compound generated in lignin degradation: formaldehyde. Formaldehyde is a small aliphatic aldehyde that is often overlooked in bioprocessing. Yet it is inhibitory to ethanol-generating yeast at concentrations as low as 1.0 mM, lower than is found in many chemically pretreated lignocellulosic feedstocks, resulting in a reduction in product generation of up to tenfold [18]; formaldehyde accumulation is often a challenge overcome via engineered resistance [19]. It can be inhibitory even to the organisms that generate it: formaldehyde detoxification can prove a rate-limiting step in the microbial degradation of methoxylated aromatic compounds [8,9,20]. In this study, we have taken advantage of the single-carbon (C_1_) metabolism of the model methylotroph *Methylorubrum extorquens* (formerly *Methylobacterium extorquens*), which uses formaldehyde as a central metabolic intermediate [21]. Our aim was to investigate the potential for *M. extorquens*, as a member of a defined lignocellulose-degrading community, to increase the efficiency of lignocellulosic breakdown by consuming the inhibitory formaldehyde. 

The *Methylobacterium* and *Methylorubrum* clade encompasses a diverse range of species in a variety of metabolic niches, including commensal relationships with plants as well as independently, in soil and leaf litter [22]. While the extent of the metabolic capabilities of the genus is not fully characterized—and recent work suggests some species may have the ability to utilize lignin-derived aromatic compounds [23]—we chose to include in our consortium, *M. extorquens* PA1, a model organism for which extensive metabolic and physiological data exist [21,24,25,26]. The other bacterial members of this defined community included *Pseudomonas putida*, a canonical lignin degrader that has been studied extensively for its aromatic catabolism [20,27,28,29,30], and *Cellulomonas fimi*, a cellulose degrader of interest for its ability to utilize diverse polysaccharides and to channel the products of their degradation to other organisms in co-culture [31,32,33]. In some experiments, a fourth microbial strain, the oleaginous yeast *Yarrowia lipolytica*, was included for its ability to grow on organic acids generated by other consortium members, and to produce neutral lipids as a potential end product [33,34,35]. We envision, ultimately, developing a stirred aerobic bioreactor for the transformation of lignocellulose hydrolysate; for this reason, we chose not to work with mycelial fungi or filamentous bacteria. 

In place of a complex and undefined plant biomass substrate, we opted for a simple and defined set of compounds to stand in for lignocellulose: cellobiose (a disaccharide of glucose, a product of cellulose hydrolysis), xylose (a 5-carbon sugar found in hemicellulose), and vanillic acid (a simple methoxylated aromatic compound, a derivative of guaiacyl lignin phenylpropanoids, for which formaldehyde generation is the first step in catabolism by *P. putida* [20]). Of these compounds, *P. putida* could consume only vanillic acid and *C. fimi* only cellobiose and xylose (with cellobiose the preferred substrate [33]); *M. extorquens* and *Y. lipolytica* could consume neither and, therefore, subsisted on metabolites generated by *P. putida* and *C. fimi*. The hypothesized interactions around which this community is built are shown in Figure 1.

Our ultimate goal is the development of a metabolically efficient and ecologically robust model microbial lignocellulose-degrading community, in which we can take advantage of recent developments in metabolomic measurement and metabolic modeling to enable a flexible, predictive strategy to maximize community output [36]. To achieve this, we sought to establish robust and reproducible methods for the culture of this novel community and for measurement of organism growth, activity, and interactions; to characterize the dynamics of formaldehyde during the consumption of lignin-derived aromatic compounds.

## 2. Materials and Methods

Specifics of the strains and culture conditions used in each experiment are given in Appendix A. Details are provided below; abbreviations used in Appendix A are denoted by square brackets.

*Strains*: The strains used were *Cellulomonas fimi* ATCC 484 [CF], *Pseudomonas putida* KT2440 [PP], *Yarrowia lipolytica* CLIB122 [YL], and three strains of *Methylorubrum extorquens* PA1 [ME]. Strain CM2730 is the wild-type strain, *M. extorquens* PA1 Δ*celABC*: the cellulose-synthesis locus was deleted to eliminate clumping and facilitate growth analysis by OD [26]. Strain CM3745 is *M. extorquens* PA1 Δ*celABC* Δ*efgA*, a mutant with increased tolerance to formaldehyde [37]. CM3745 was used in early experiments, but as the formaldehyde concentrations in coculture never exceeded the maximum inhibitory concentration (MIC) of wild-type *M. extorquens* and we observed no difference in performance between the two strains, later experiments were conducted with CM2730. 

Strain CM4744 was a methionine-overproducing strain developed for this study, by taking advantage of the fact that methionine overproduction can confer resistance to the methionine-analog ethionine [38,39]. *M. extorquens* CM2730 was grown to stationary phase in MP + 15 mM methanol. A total of 200 µL of culture was spread-plated onto each of two plates with MP-methanol agar medium containing 1 mg/mL of ethionine. A total of 8 colonies were chosen and re-streaked onto fresh MP-methanol-ethionine plates. These isolates were then tested for their ability to promote the growth of *C. fimi* in the absence of methionine, and all performed equally well. Results shown in this text (Figure 9) are from a single isolate, CM4744.

For all experiments, freezer stocks were streaked onto MPI agar plates with 15 mM succinate or 125 mM methanol (*M. extorquens*), or onto Nutrient Agar (Difco) (other strains), to obtain colonies. A single colony was used to inoculate 5 mL of liquid culture medium and grown overnight until stationary phase. Unless otherwise noted, pre-growth medium was MP with methionine/thiamine/biotin supplements in standard concentrations and 15 mM methanol (*M. extorquens*) or 10 mM glucose (other strains). When necessary, this inoculum was subcultured once into a different medium (e.g., Model Lignocellulose) for an additional 24 h of growth to acclimate it for the experiment. After 24 h, stationary-phase cultures of all species were diluted to normalize their ODs to match that of the least-dense culture, then equal volumes of all cultures were inoculated into the experimental medium, at a dilution of 1:64 (vol:vol) into the final medium, unless otherwise stated.

*Basal medium and buffer*: The medium used as a basis for all growth experiments was a modified PIPES-buffered medium previously described [26] [MP]: 30 mM PIPES, 1.45 mM K_2_HPO_4_, 1.88 mM NaH_2_PO_4_, 0.5 mM MgCl_2_, 5.0 mM (NH_4_)_2_SO_4_, 0.02 mM CaCl_2_, 45.3 µM Na_3_C_6_H_5_O_7_, 1.2 µM ZnSO_4_, 1.02 µM MnCl_2_, 17.8 µM FeSO_4_, 2 µM (NH_4_)_6_Mo_7_O_24_, 1 µM CuSO_4_, 2 µM CoCl_2_, 0.338 µM Na_2_WO_4_, pH 6.7. As described in the Results and in Table 1, we conducted some experiments with a variant of MP with lower PIPES or higher phosphate [P] concentrations. “P” refers to the combination of K_2_HPO_4_ and NaH_2_PO_4_ in the same ratio as in the final medium. Most experiments used either 1× (30 mM) or 0.1× (3 mM) PIPES, and either 1× P (~3 mM phosphate) or 5× P (~15 mM phosphate). 

*Carbon substrates and supplements*: Carbon substrates consisted of glucose [G], methanol [MeOH], succinate [S], vanillic acid [VA], protocatechuic acid [PCA], cellobiose [CB], and xylose [XY]. All were stored as sterile aqueous stock solutions and added to cultures by pipet. Because of their poor solubility in water, VA and PCA were stored as 50 mM stock in MP medium so that their addition would not dilute the final culture medium. Whereas most carbon substrates were sterilized by autoclaving, VA and PCA were sterilized by filtration out of an abundance of caution because we found that autoclaving changed their color (brown and purple, respectively). 

For the experiments identifying the nutritional needs of *C. fimi* (Appendix A), all amino acid stocks were made in MP medium at 10 g/L (10 mL vol), except asparagine (6.67 g/L), aspartic acid (2 g/L, NaOH added to bring pH to 6.5), and tyrosine (50 g/L in DMSO), due to solubility. For all stocks, regardless of concentration, 100 µL of stock added to 10 mL culture medium, with the exception of the tyrosine DMSO stock, of which only 10 µL was added. Wolfe’s vitamins made according to [40] and provided at 1×. Yeast extract (VWR) was provided at a final concentration of 0.3 g/L and tryptone (Peptone from Casein, Sigma Aldrich) at 0.16 g/L.

In Model Lignocelluose medium, methionine [M], thiamine [T], and biotin [B] aqueous stocks were provided at concentrations of 2 mg/L, 5 µg/L, and 40 µg/L, respectively (standard concentrations), unless otherwise noted. Stocks were made in water, filter-sterilized, stored at 4 °C, and added to medium prior to each experiment. For experiments testing vanillic acid toxicity, glucose was provided for *C. fimi* and methanol for *M. extorquens* as additional carbon substrates as they cannot grow on vanillic acid, but no additional carbon substrate was provided for *P. putida*. For experiments testing the effect of iron concentration, 17 mM (1000×) aqueous, autoclave-sterile FeSO_4_ stock was added.

For use in enumerating colony-forming units of the different species from cocultures, standard MP medium was prepared with 1.8 g/L glucose and 4.05 g/L sodium succinate dibasic hexahydrate, methionine/thiamine/biotin in standard concentrations, and 15 g/L agar. 

Formaldehyde was produced fresh weekly as 1 M stock by combining 0.3 g paraformaldehyde powder (Sigma Aldrich, St. Louis, MO, USA), 9.95 mL ultrapure water, and 50 µL 10 N NaOH solution in a sealed tube and immersing in a boiling water bath for 20 min to depolymerize. The stock was stored at room temperature and removed from the sealed tube using a syringe when needed.

*Vessels and incubation conditions*: All cultures were grown at 30 °C, either in culture flasks, culture tubes, multiwell culture plates, or culture plates with solid medium [agar]. For multiwell culture plates [multiwell], we used 48-well tissue culture plates (Corning Costar, Tewksbury, MA, USA) with a total volume of 640 µL per well, and incubated in a LPX44 Plate Hotel (LiCONiC, Mauren, Liechtenstein) with shaking at 650 RPM. For glass tubes, we originally used Balch tubes with serum stoppers [Balch] (Chemglass, Vineland, NJ, USA) in order to ensure that volatile compounds such as formaldehyde were not lost in the gas phase; however, experiments demonstrated no difference in any measured compounds between Balch tubes and simple 16 × 150 mm glass culture tubes with loose-fitting lids [tube], so we ultimately used aerobic tubes for convenience. All culture tubes contained 5 mL of liquid medium unless otherwise stated and were incubated with shaking at 250 rpm. For culture flasks [flask], we used 50 mL-capacity glass Erlenmeyer flasks, containing 10 mL of liquid culture, also shaken at 250 rpm.

*Measurements*: In growth experiments conducted in tubes or flasks with multiple timepoints, a typical sampling procedure was as follows: 100 µL of culture was removed from the vessel, by syringe and needle through the stopper for Balch tubes or by pipet otherwise, transferred into a microcentrifuge tube, and centrifuged for 1 min at 14,000× *g* to pellet the cells. 60 µL of supernatant was used for formaldehyde measurement and 20 µL for GC-MS or HPLC analysis. The cell pellet was reserved for species-specific analysis of cell abundance as colony-forming units [CFU]. Not all analyses were carried out for all timepoints, but the sampling procedure nonetheless remained the same. Abbreviations in Appendix A are as follows: optical density at 600 nm [OD]; formaldehyde [F]; cellobiose by HPLC [HPLC]; vanillic acid or protocatechuic acid by GC-MS [GC-MS].

For CFU measurement, cell pellets were resuspended in 980 µL of MP medium without carbon substrate (1:10 dilution) and then subjected to serial 1:10 dilutions down to 10^−^^6^ (7 dilutions total). These dilutions were either spread-plated or spot-plated onto solid culture medium. For spot-plates, three replicates spots of 10 µL of each dilution were pipetted onto plates and spots were dried under a laminar flow hood, then incubated at 30 °C for 4 days or until colonies were visible. Species were identified by colony morphology (Appendix A). The colonies in each series of 7 dilution spots was counted; the number of colonies in the two spots of the highest dilution levels that had countable colonies were summed, then multiplied by 1.1 times the lower of the two dilution factors to calculate the number of colony-forming units (CFU) per mL in the original undiluted sample. The mean and standard deviation were calculated for the three replicate spot series representing each sample. For spread-plates, between 1 and 3 dilutions were chosen for plating based on predicted cell abundance; from each dilution, 100 µL of the dilution was spread onto a culture plate. Plates were dried and incubated as for spot-plates. CFU/mL was calculated by multiplying by the dilution factor and accounting for the volume plated.

Measurement of optical density at 600 nm (OD_600_) for cultures in glass tubes was carried out nondestructively by reading the whole tube with a Spectronic 200 spectrophotometer (Thermo Fisher, Waltham, MA, USA). For cultures in flasks, a 100 µL sample was transferred to a trUVue low-volume cuvette (Bio-Rad, Hercules, CA, USA) and read in a SmartSpec Plus spectrophotometer (Bio-Rad). For experiments in multiwell plates, optical density was assessed using a Wallac 1420 Victor2 Microplate Reader (Perkin Elmer, Waltham, MA, USA), reading OD_600_ for 0.4 s. In experiments involving different carbon substrates, blank wells were included for each medium composition for blanking purposes, as solutions containing PCA and VA were purple and brown, respectively.

Formaldehyde was measured using the method of Nash [41]. Reagent B was prepared as described (2 M ammonium acetate, 50 mM glacial acetic acid, 20 mM acetylacetone); for each assay, equal volumes of sample (or standard) and Reagent B were combined in a microcentrifuge tube and incubated for 6 min at 60 °C. Absorbance was read on a spectrophotometer at 412 nm and formaldehyde concentration calculated using a standard curve made from freshly prepared formaldehyde stock. To assay large numbers of samples, a 96-well polystyrene flat-bottom culture plate (Olympus Plastics, San Diego, CA, USA) was used, with a total volume of 200 µL per well and incubation time of 10 min before absorbance at 432 nm was read using a Wallac 1420 Victor2 plate reader. A clean plate was used for each assay; each plate contained each sample in triplicate as well as a standard curve run in triplicate. The absorbance of vanillic acid was not found to interfere with formaldehyde measurements.

Vanillic acid was measured by gas chromatography-mass spectrometry (GC-MS) using an extraction and derivatization procedure modified from [42]. 20 µL of culture supernatant was combined with 1.2 µL of 1 M HCl to acidify to pH ~ 2. The sample was combined with 100 µL of a 1:100 mixture of 2-chlorobenzoic acid:ethyl acetate, and vortexed to extract the vanillic acid into the organic phase. The sample was centrifuged at 14,000× *g* for 1 min to separate the phases, and 80 µL of the organic phase was transferred to a clean GCMS sample vial (1 mL capacity). Samples were dried in a fume hood, then 400 µL of derivatization reagent was added. The derivatization reagent consisted of a 99:1:1000 mixture of N,O-Bistrifluoroacetamide:Trimethylsylil chloride:acetonitrile (that is, BSTFA-TMCS (TCI America, Portland, OR, USA) diluted 1:10 in acetonitrile). The sample was incubated, sealed, at 70 °C for 30 min, then cooled. Samples were analyzed on a Shimadzu GCMS-QP2010 Plus with a 30 m × 0.25 mm dimethyl polysiloxane column (Rxi-1ms, Restek, Bellefonte, PA, USA) in splitless mode with a 1-min injection at 280 °C. The GC run program was as follows: hold at 80 °C for 1 min; ramp to 110 °C at 20 degrees/min; ramp to 240 °C at 10 degrees/min; ramp to 280 °C at 40 degrees/min; hold at 280 °C for 5 min. The MS was run on SIM mode. Vanillic acid was detected as as 3-methoxy-4-[(trimethylsilyl)oxy]-benzoic acid trimethylsilyl ester, with retention time 10.4 min. Characteristic fragments used for quantitation were at *m/z* 267 and 297, with other fragments at 126, 193, 253, 312. Standard curves were generated from vanillic acid stocks made in lab and extracted alongside the samples.

Cellobiose and xylose were measured using a Shimadzu LC-20 high-performance liquid chromatograph (HPLC). Supernatant samples were filtered through a 0.2 µm syringe filter to remove any particles, and diluted in water if necessary to maintain signal within quantifiable range. They were run on an Aminex HPX-87h column (Bio-Rad) at a flow rate of 0.6 mL/min, column temperature 30 °C, with 5 mM H_2_SO_4_ as eluent. Peaks were detected using a RID-20A refractive index detector: cellobiose with a retention time of 7.1 min and xylose at 9.3 min. 

*Formaldehyde tolerance distributions*: The distribution of formaldehyde tolerance phenotypes within a population was assessed by counting colony-forming units on agar medium containing formaldehyde, as described previously [43]. MP medium was prepared with the necessary carbon substrates and supplements for each species; after autoclaving, the medium was cooled to 50 °C and formaldehyde was rapidly mixed in. Agar was poured into 100 mm culture plates, dish lids were replaced, and plates were allowed to cool on the benchtop. Plates were stored at 4 °C for no longer than 1 week. Cultures were grown to stationary phase on MP medium with a preferred carbon source (methanol for *M. extorquens*; glucose for other species), then CFU were spot-plated as described above onto a series of plates containing a range of formaldehyde concentrations, and the number of cells capable of forming colonies at each concentration of formaldehyde was calculated. Note that an abundance of 34 CFU/mL is necessary to observe 1 cell per 30 µL plated, so for a population of ~2 × 10^8^ CFU/mL (as was typical for *M. extorquens* samples), this method has a limit of detection of 1.65 × 10^−^^7^.

*Spent medium experiment*: To generate *P. putida* spent medium, *P. putida* was grown on Model Lignocellulose medium (Table 1) to stationary phase. The culture was then centrifuged and the supernatant filtered through a 0.2 µm filter to remove cells. Vanillic acid was added again to a final concentration of 4 mM, to replenish the vanillic acid consumed by *P. putida*. This was then used as the growth medium for *C. fimi*.

*Data analysis and visualization*: Original data are available as spreadsheets in Appendix A. All data were analyzed using R v. 4.0.2 in Rstudio v.1.3.959. Growth rates (*r*) were calculated by fitting the exponential portion of the growth curve to the model *N*(*t*) = *N*_0_*e^rt^*. Lag time was calculated as the intersection of the fitted growth curve with OD = 0.0126 (the threshold of detection in multiwell plates). The relationships between lag time and substrate concentration, or between growth rate and concentration, were calculated by fitting a linear relationship using the lm package in R. Inkscape v. 1.0 was used to generate the conceptual figures (Figure 1 and Figure 10) and for customizing layout and annotations on other figures. 

## 3. Results

### 3.1. Both Vanillic Acid and the Formaldehyde Generated during Its Degradation Are Toxic to Consortium Members

Our initial experiments aimed to define the role of toxic compounds in our community, particularly formaldehyde generation and consumption, and formaldehyde-mediated growth inhibition. Previous work in our lab has shown that *P. putida* growing on vanillic acid can generate formaldehyde that escapes the cell to accumulate in the growth medium [23]. However, the effect of that formaldehyde on other members of the microbial community is unknown. Moreover, given that lignin-derived aromatic compounds can themselves be toxic, we reasoned it was possible that vanillic acid could also inhibit growth in our consortium and that the benefit of its degradation might even outweigh the risks of formaldehyde production. We, therefore, assessed the tolerance of consortium members to both compounds.

Because *P. putida* can grow on vanillic acid as a sole carbon source and would be required to do so in our consortium, we tested its growth tolerance to a range of vanillic acid concentrations while using the compound as its sole carbon substrate. For the other organisms, we were required to provide an alternative growth substrate in addition to the vanillic acid (see Appendix A for details). Remarkably, vanillic acid had a very small effect on the growth rate (slope = −0.0032, *p* < 0.05), and no detectable effect on the lag time, for *M. extorquens*, up to 14 mM (Figure 2). However, vanillic acid did inhibit the growth of both *C. fimi* and *P. putida*; both organisms showed a significant increase in lag time (slope = 1.77 and 0.93, respectively; *p* < 0.05), and *C. fimi* showed a decrease in growth rate (slope = −0.0097, *p* < 0.05), with increasing vanillic acid concentrations (Figure 2, Appendix A). For *C. fimi*, no growth was detectable at 14 mM vanillic acid and it was not possible to calculate lag time above 8 mM.

For *P. putida*, we conducted similar experiments on protocatechuic acid, another lignin-derived aromatic compound that lacks the methoxyl group but is otherwise identical to vanillic acid. While *P. putida* showed a slight decrease in growth rate with PCA concentration, it was much lower than that for vanillic acid, and there was no detectable change in lag time (Appendix A). It is therefore likely that much of the toxic effect on *P. putida* is due to the methoxyl group of the vanillic acid, either directly, or due to the formaldehyde that can be generated from it.

We also measured the effect of formaldehyde on the consortium members, by assessing growth in liquid medium, and by measuring the frequency distribution of formaldehyde-tolerant individuals by counting colony-forming units on formaldehyde agar [43]. In both media, we found that *C. fimi* showed no growth at concentrations of 0.5 mM and higher, establishing it as the most formaldehyde-sensitive of the organisms in our consortium. In contrast, 100% of *M. extorquens* cells are able to grow in the presence of 1 mM formaldehyde, and both *P. putida* and *M. extorquens* showed some growth at concentrations of 3 mM or higher (Figure 3).

In our experiments with *P. putida* growing on vanillic acid as the sole carbon source, formaldehyde levels in the medium increased throughout the period of growth and decreased only when vanillic acid was depleted and the culture entered stationary phase (Figure 4). Regardless of the initial concentration of vanillic acid, the formaldehyde concentration uniformly reached a peak of between 0.6 and 0.8 mM; however, because higher concentrations cause *P. putida* to grow more slowly, formaldehyde remained present in the medium for a longer time. In contrast, growth on PCA resulted in faster growth than on vanillic acid and no formaldehyde production (Figure 4). Notably, because *C. fimi* growth is inhibited at concentrations of 0.5 mM and higher, these results pointed to the potential for lignin degradation to interfere with cellulose degradation, unless a detoxification mechanism for formaldehyde could be introduced. Furthermore, while *P. putida* showed very little inhibition from formaldehyde concentrations lower than 1 mM in our single-species experiments, we reasoned that removal of formaldehyde in the medium might also help relieve some of the burden of intracellular detoxification from *P. putida*.

### 3.2. M. extorquens Reduces the Formaldehyde Concentrations in Cocultures Growing on Vanillic Acid

We therefore tested the hypothesis that co-culturing *M. extorquens* with *P. putida* growing on vanillic acid could lower the formaldehyde concentrations in the medium, as *M. extorquens* can use formaldehyde as a growth substrate. We conducted a number of experiments at different vanillic acid concentrations and adding the two organisms at different ratios. We consistently found that including *M. extorquens* did indeed result in lower concentrations of measurable formaldehyde (Figure 4). While preliminary experiments indicated that the amount of formaldehyde reduction could be influenced by the conditions in which *M. extorquens* was grown prior to being added to the coculture (Appendix A), in the spirit of creating a robust and sustainable community that might withstand serial culture, we ultimately proceeded with experiments in which all organisms were grown to stationary phase in similar conditions before being combined.

The modest change in formaldehyde concentrations did not noticeably alter the overall growth rate or yield of the *P. putida* + *M. extorquens* coculture (Figure 4). We next needed to test whether it would have an effect on *C. fimi*, the most formaldehyde-sensitive member of the consortium. To do so required developing a set of culture conditions that would support growth of all community members together.

### 3.3. A Minimal Growth Medium Can Support All Members and Facilitates Metabolomic Analysis, with Modest Amino Acid and Vitamin Supplements and Reduced Buffer Concentrations

For experiments in the community dynamics of the consortium, we needed to develop a new culture medium that would not only support the growth of each consortium member individually, but also facilitate full chemical characterization of their interactions, and enable reproducible experiments. Because a defined mineral medium can simplify metabolomics analysis and avoid the batch effects sometimes observed in complex media, we began with a PIPES-buffered mineral medium (MP) that had originally been optimized for *M. extorquens* [26]. We found that it supported *P. putida* growth well, on multiple carbon substrates. However, *C. fimi* showed no measurable growth on MP on any carbon source. The addition of yeast extract or peptone did aid growth, leading us to test the possibility of an amino acid or vitamin auxotrophy. As most published media for *Cellulomonas* species contain undefined media or supplemental vitamins (e.g., [44,45]), we conducted set of trials using different combinations of amino acids and vitamins to deduce that *C. fimi* required supplementation by methionine and thiamine in order to grow on MP (Appendix A). Biotin was also added to the medium, due to an indication early in experimentation that it might help *C. fimi* growth, and from literature mentioning that it might be necessary for *Y. lipolytica* [46,47] and *C. fimi* [45], although in many cases its addition did not noticeably affect growth of any of the organisms (Figure 5). The addition of these supplements had no measurable effect on the growth of *M. extorquens* and only a very small effect on *P. putida* (Figure 5).

During the single-species growth assays we also measured growth rates of each of the organisms alone in MP medium, in optimal growth conditions (Appendix A). Growth rates differed markedly among organisms: *P. putida* grew most rapidly by far (*r* = 0.40), *C. fimi* the slowest (*r* = 0.21), and *M. extorquens* and *Y. lipolytica* with similarly moderate growth rates (*r* = 0.24 and 0.25, respectively). While these differences posed a challenge in terms of our understanding of metabolic interactions among consortium members, it ultimately made it relatively easy to interpret growth curves of mixed cultures (as described further below).

A final amendment to the growth medium recipe resulted from some preliminary work we conducted testing the feasibility of untargeted metabolomics analysis on the community. It was found that the 30 mM PIPES buffer in the medium interfered with sample processing for LCMS and GC-MS analysis. To remedy this problem, we explored the possibility of lowering the buffer concentration and explored the effect of this on the growth of the consortium members. We found that lowering the PIPES concentration had very little effect on the growth of most organisms, but *P. putida* proved the exception. However, it was still able to maintain reasonable growth at a PIPES concentration of 3 mM (tenfold lower than the original) (Appendix A). Supplementation of extra phosphate aided growth as well, though phosphate also interferes with metabolomics analysis, so some experiments were conducted with 5× the original phosphate concentration, but not all. 

As mentioned above, we chose as carbon substrates a set of relatively simple compounds with chemical similarity to the key components of lignocellulose: cellobiose, xylose, and vanillic acid (Figure 1, Table 1). When necessary, we used 15 mM methanol or 3.5 mM succinate to support the growth of *M. extorquens*, though in many experiments we chose not to supplement *M. extorquens* growth beyond the formaldehyde and organic acid generated by the other consortium members. To decide upon the concentration of each lignocellulose-derived compound to include, we considered the typical balance among lignin, cellulose, and 5-carbon sugar components found in plant-based feedstock [48] and chose a concentration high enough to support measurable microbial growth within an experiment lasting 1–3 days, while being low enough to mitigate the effects of vanillic acid toxicity. We began with the 4:5:4 molar ratio of cellobiose:xylose:vanillic acid listed in Table 1, which translates to a 46:24:30 ratio by C molarity or a 49:27:24 ratio by mass. We then tested different concentrations. *C. fimi* grew fastest on the most dilute medium; at the intermediate concentration it would likely have achieved a higher final yield but did not do so within the length of the experiment (60 h); at the highest concentration it showed no growth at all. *P. putida* also showed earlier growth at the lower carbon concentrations (time to reach OD ~0.6 on 1× C was approximately 2 h earlier than on 3×), but due to its faster growth rate and moderate tolerance to vanillic acid, it recovered rapidly from the toxic effects of the concentrated medium and ultimately achieved the highest final optical density in the medium with the most carbon (Figure 6). The final recipe for our basal growth medium, which we called Model Lignocellulose, is given in Table 1. Details on the individual growth conditions of each experiment described in this manuscript are given in Appendix A. 

We developed methods to measure the production and consumption of all the compounds of interest: a colorimetric method for formaldehyde, GC-MS for vanillic acid, and HPLC for cellobiose and xylose. We also explored ways to measure the dynamics of individual species as they grew together in liquid co-culture. While we were unable to distinguish among species using flow cytometry, they were easily distinguished by colony morphology when plated onto agar medium (Appendix A). Furthermore, because *C. fimi*, *M. extorquens*, and *P. putida* had such distinctive growth rates, each formed a different part of the growth curve, and it was, therefore, possible to make an initial qualitative assessment of culture dynamics based on optical density alone. In general, *P. putida* was recognized as the earliest, most rapid part of the growth curve, which was followed by a slight dip in OD once *P. putida* had exhausted its carbon substrate (likely due to accumulation we observed of *P. putida* cells on the walls of the culture vessel, or to change in cell shape as has been observed for *M. extorquens* [26]). *C. fimi* growth, being much slower, was recognized as the slower growth becoming prominent well after *P. putida* reached its maximum. With its moderate growth rate, *M. extorquens* was recognizable as any increase in the growth curve relative to the *C. fimi* growth alone—though some of that increase in OD may also have been attributable to a stimulation of *C. fimi* growth by *M. extorquens*. An example of these dynamics is visible in Figure 7.

### 3.4. P. putida Inhibits Growth of C. fimi in Model Lignocellulose Medium, and C. fimi Supports the Growth of M. extorquens

Having developed a medium in which we could grow all organisms in coculture, we next set out to measure the dynamics of formaldehyde production and consumption on the microbial community, specifically to address the question of whether formaldehyde production by *P. putida* had an adverse effect on the activity of *C. fimi* (the most formaldehyde-sensitive of the organisms), and whether the presence of *M. extorquens* could modulate that effect. While the pulse of formaldehyde produced by *P. putida* during growth on vanillic acid was above the maximum inhibitory concentration (MIC) of *C. fimi*, it was transient and, therefore, we did not know how to predict its effects.

We conducted several experiments in which we observed members of the consortium alone or in different combinations, growing in Model Lignocellulose medium or similar conditions (Figure 7 and Appendix A). We consistently found that while *C. fimi* (with or without *M. extorquens*) was able to reach a high OD after 60–80 h of growth, when we added *P. putida* to the culture, that high OD was never reached. This was true in medium with 4 mM vanillic acid, in which we observed an early peak in OD at <20 h due to *P. putida* growth, and in medium with only 2 mM vanillic acid, in which much less *P. putida* growth was supported and yet *C. fimi* growth was still inhibited. The inhibited growth was also reflected in slower cellobiose consumption by *C. fimi* and in reduced final colony counts (Figure 7). 

From the colony counts, we also observed that the presence of *C. fimi* supported the growth of *M. extorquens*, even when no carbon substrate was provided for *M. extorquens*, in agreement with our initial hypothesis that *C. fimi* might produce substrates upon which *M. extorquens* could cross-feed. Yet when *P. putida* was present, *M. extorquens* abundance was depressed; this may have been an indirect effect due to the reduced activity by *C. fimi*, or a direct interaction between *P. putida* and *M. extorquens* (Figure 7).

In cultures with *P. putida*, we observed the expected peak in formaldehyde production between 10 and 20 h of growth; as observed previously, the addition of *M. extorquens* to the coculture reduced the magnitude and duration of this peak (Figure 7 and Appendix A). Furthermore, the addition of *M. extorquens* resulted in a slight increase in OD relative to a coculture with only *P. putida* and *C. fimi*. However, *M. extorquens* was not able to completely ameliorate either the formaldehyde production or the growth inhibition. This observation led us to investigate whether the production of formaldehyde was in fact the primary reason for the effect of *P. putida* on *C. fimi*. If it was, then improving the performance of *M. extorquens* could dramatically improve the efficacy of the consortium in degrading our model lignocellulose compounds. 

### 3.5. P. putida Inhibition of C. fimi Growth May Be due to Multiple Mechanisms

To test our hypothesis that formaldehyde production was the reason for the effect of *P. putida* on *C. fimi* growth, we tested growth on a modified lignocellulose medium, in which we replaced vanillic acid with protocatechuic acid (PCA). PCA is the immediate product of vanillic acid demethoxylation, and, therefore, can serve as a control for *P. putida* growth on aromatic compounds without formaldehyde production. We found that *P. putida* also inhibited *C. fimi* in these formaldehyde-free growth conditions, in a manner consistent with our observations on vanillic acid and independent of the addition of *M. extorquens*, leading us to conclude that formaldehyde production was not the most important aspect of the interaction between *C. fimi* and *P. putida* (Figure 8A).

We next explored several other hypotheses to explain the inhibitory effect of *P. putida* on *C. fimi*. To assess whether *P. putida* was competing with *C. fimi* for important compounds in the medium, we tested higher concentrations of methionine, thiamine and biotin, or iron. In case the inhibition was due to an excreted compound, we also tested whether spent medium from *P. putida* could inhibit *C. fimi* growth. We found evidence supporting all three potential mechanisms. Addition of iron and vitamin/amino acid supplements, both alone and in combination, improved growth of the coculture (Figure 8). This was true only to a point; the highest iron concentrations tested resulted in no growth, likely due to toxic effects. Whereas the addition of iron resulted in an increase in the growth rate during the time <15 h, indicating an effect on *P. putida* growth, the effect of methionine, biotin, and thiamine was seen primarily in the second portion of the growth curve (*C. fimi* and *M. extorquens*). Thus, although our observations from pure culture indicate that thiamine and biotin provide *P. putida* with only a slight improvement in growth and methionine has no measurable effect, the species might still be actively removing these compounds from the medium. 

Moreover, while the addition of *P. putida* spent medium did not affect the initial growth rate of a *C. fimi*-*M. extorquens* coculture, it did reduce the final yield (Figure 8). This could indicate either the effect of a soluble compound produced by *P. putida*, or the depletion by *P. putida* of important medium components, or both. One possibility might be siderophores, as *P. putida* produces siderophores that can inhibit the growth of competitors, and has been observed to accelerate their production in response to the availability of aromatic growth substrates [49,50]. 

Because increasing the iron concentration in the medium benefitted the growth of the consortium, we considered changing the base recipe for our model lignocellulose medium. However, we found that adding high concentrations of iron was in fact inhibitory to *C. fimi* growth when *P. putida* was not present to consume it (Appendix A). Clearly, the ideal growth medium to support interactions among the members of the consortium was not the same as the medium that best supported each organism individually.

### 3.6. Methionine-Overproducing M. extorquens Can Support the Growth of C. fimi

Given our observation that adding supplements to the medium could create unintended consequences, we opted to explore the possibility of including a consortium member with the ability to provide the methionine required by *C. fimi* for growth. This would not only eliminate the necessity for us to add it to the medium, but also create the opportunity for positive feedback between *C. fimi* and *M. extorquens*, given our data suggesting that *C. fimi* could support *M. extorquens* growth.

We used a previously published method to generate a methionine-overproducing strain of *M. extorquens* by selecting on medium containing ethionine [38,39]. When we cultured this *M. extorquens* strain with *C. fimi,* or both *C. fimi* and *P. putida,* in medium without added methionine, we found that it was able to support the growth of *C. fimi* to abundances similar to or better than those supported by 20 mg/L methionine (Figure 9). This effect was unique to the overproducing strain: wild-type *M. extorquens* did not promote *C. fimi* growth. Moreover, the effect was dependent upon the presence of either methanol or succinate to support the growth of the methionine producer (Figure 9), indicating that the benefit required having a high abundance of growing *M. extorquens* cells. The inclusion of a methionine-overproducing strain in the consortium might therefore prove to be a remedy to resolve one of the several interactions between *C. fimi* and *P. putida*—the competition for methionine.

## 4. Discussion

We have established a system for studying microbial lignocellulose degradation that uses tractable, well-characterized microbial species in ecological roles that take advantage of their evolved metabolic capabilities, and have developed a simple defined Model Lignocellulose medium with which to investigate interspecies interactions. This groundwork will enable future investigations involving metabolomic analysis and modeling for a more complete understanding of metabolic exchange within the community, and further work using engineering and laboratory evolution to optimize the efficiency and yield of the biomass transformation. While much remains to be learned about the mechanisms of the dynamics observed in this microbial community, the methods and initial findings described here comprise the first step toward using this promising model consortium for studying the microbial transformation of lignocellulosic biomass.

A valuable lesson learned from this study relates to the prevalence of unexpected ecological interactions that may be discovered even in a simple three-species community (Figure 10). Dynamics we discovered that were not originally predicted included the dramatic interspecies differences in nutritional needs, individual growth rates, pH sensitivity, and tolerance to toxins such as formaldehyde and vanillic acid. Some of these differences may provide clues about the organisms’ ecological niches. For example, formaldehyde tolerance shows a link to species’ metabolic capabilities: in *M. extorquens*, a methylotroph, formaldehyde is detoxified by the dephospho-tetrahydromethanopterin pathway [51,52], and vanillic-acid-consuming *P. putida* carries a set of redundant glutathione-dependent formaldehyde dehydrogenases [30]. However, in *C. fimi*, to our knowledge, formaldehyde detoxification has not been characterized. Another complication in our understanding of the community’s dynamics was the consumption by *P. putida* of compounds (iron and methionine) seemingly in excess of what was limiting for its growth. Critically, very little about these findings would be captured by genome-scale metabolic modeling, a method often based on the assumption that each member achieves optimal metabolism on the available substrates [53]. Our results point to the importance of taking into account traits outside of metabolism for their roles in interspecies interactions. As many model experimental systems in microbial ecology focus primarily on metabolite exchange and chemical warfare (e.g., [38,54,55,56]), our microbial consortium could prove a useful model for the study of alternative modes of ecological interactions.

The discovery of *C. fimi*’s dependence on methionine for growth provided a fortuitous opportunity to enrich the existing interspecies interactions though the development of a methionine-excreting strain of *M. extorquens* (Figure 9). This is especially promising, as we have observed *C. fimi* already has the capacity to promote *M. extorquens* growth (Figure 7), likely through the generation of carbon substrates. Prior work using exometabolomic analysis of *C. fimi* growing on cellulose and galactomannan has shown it to produce several organic acids (e.g., alpha-ketoglutaric acid, 3-hydroxypropionic acid, D-malic acid, citric acid, and amino acids) that are among those known to support the growth of *M. extorquens* [33]. The exchange of methionine for carbon substrates by cross-feeding is, in fact, the basis of another well-characterized model microbial consortium, which was developed through laboratory evolution and has been studied extensively as an example of a stable bidirectional costly mutualism [38,57,58]. It is likely that the growth promotion observed here could in the future be similarly developed through laboratory evolution into a true codependence.

A central goal of our work was to address the issue of formaldehyde production during lignin degradation and to test the hypothesis that a methylotrophic organism could improve the efficiency of lignocellulose degradation by removing a toxic burden. We made significant contributions in that area, by documenting formaldehyde accumulation due to *P. putida* growth on vanillic acid over time and the effect of formaldehyde consumption by *M. extorquens*. However, much remains still to be investigated. Because formaldehyde concentrations in the consortium are dynamic, their effect may be dramatically different in different growth conditions. Formaldehyde-induced cell damage occurs over time [43], and as formaldehyde remains in the medium as long as *P. putida* is actively consuming vanillic acid, a continuous-culture growth environment might induce a more substantial effect of formaldehyde on *C. fimi* viability. Yet it was clear in the present study that other interspecies interactions had a stronger effect on the community than formaldehyde, and those interactions must be resolved first before the role of formaldehyde can be accurately explored and quantified.

## Figures and Tables

**Figure 1 microorganisms-09-00321-f001:**
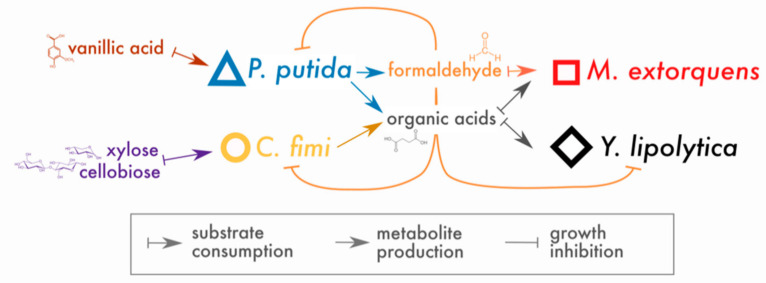
Conceptual model of interactions in the lignocellulose-degrading microbial consortium. Arrows indicate hypothesized interactions as described in the key. The colors and symbols used for compounds and species in this figure are the same as those used in data plots throughout the manuscript.

**Figure 2 microorganisms-09-00321-f002:**
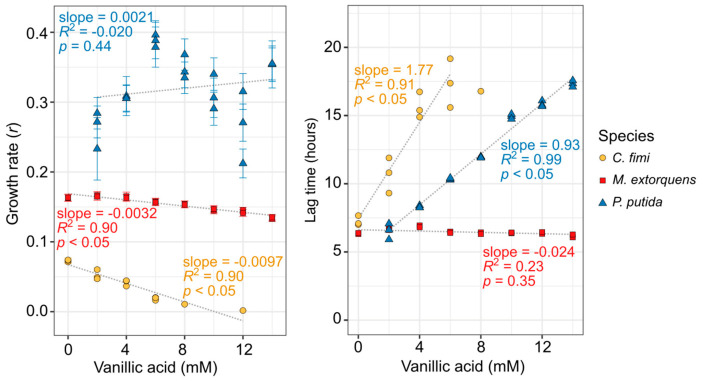
Vanillic acid inhibits the growth of *C. fimi* and *P. putida* substantially, but has only a minor effect on *M. extorquens* growth. Each organism was grown in pure culture in mineral medium and OD_600_ monitored for up to 60 h with a range of vanillic acid concentrations (original growth curves are shown in Appendix A). *C. fimi* was provided with cellobiose and *M. extorquens* with methanol as growth substrates, whereas *P. putida* was able to use the vanillic acid as a growth substrate. Each point represents a biological replicate. Only growth rates for which R^2^ > 0.9 are shown here. In the left panel, error bars denote the standard error of the fitted growth rate. For *C. fimi* and *P. putida*, lag time and growth rate are dependent on vanillic acid concentration.

**Figure 3 microorganisms-09-00321-f003:**
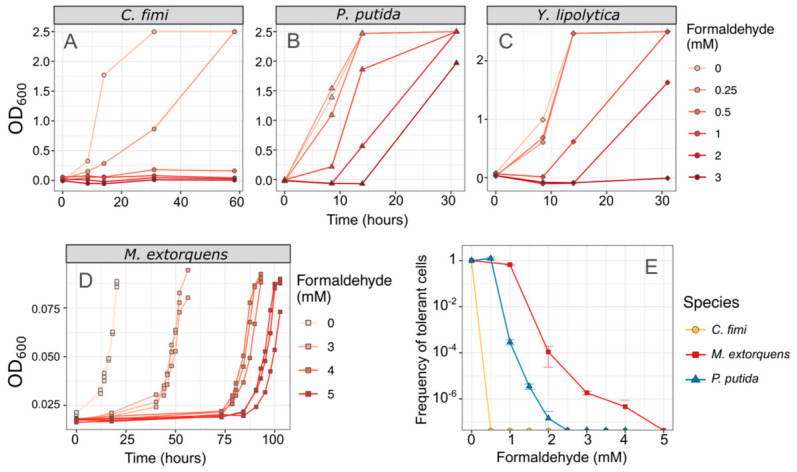
*C. fimi* is the most formaldehyde-sensitive member of the consortium. (**A**–**D**) Each species was grown in pure culture in minimal medium with varying concentrations of formaldehyde, and growth monitored by OD_600_. No growth was observed in *C. fimi* at concentrations of 0.5 mM or higher, whereas *Y. lipolytica* grew at up to 2 mM formaldehyde and *P. putida* at up to 3 mM within 30 h. Note that OD is shown here on a linear scale for ease of interpretability; note also that the color scale for formaldehyde concentrations is different between panels (**A**–**C**) and panel (**D**). Data from panel (**D**) are reproduced from [43]. (**E**) An alternative method of understanding formaldehyde tolerance is the enumeration of cells that are able to form colonies on agar medium containing formaldehyde. In *M. extorquens*, 100% of the plated population formed colonies at 1 mM and 1/10,000 formed colonies at 2 mM. *P. putida* cells also formed colonies at those concentrations but at a lower frequency. In *C. fimi*, no colonies were observed at 0.5 mM formaldehyde or higher. Error bars show the standard deviation of three replicate platings.

**Figure 4 microorganisms-09-00321-f004:**
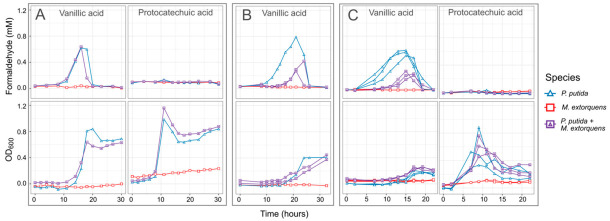
When *P. putida* grows on vanillic acid as a sole carbon substrate, formaldehyde accumulates in the medium to levels potentially toxic to *C. fimi*. When grown in coculture with *P. putida*, *M. extorquens* reduces formaldehyde accumulation. Shown here are the formaldehyde in the medium (top row) and total growth of the community (bottom row) from three separate experiments in which *M. extorquens* and *P. putida* were grown separately or together in minimal medium with vanillic acid as the sole carbon source: (**A**,**B**) 4 mM vanillic acid or protocatechuic acid; (**C**) 10 mM. Individual lines represent independent culture vessels. As *M. extorquens* cannot grow on vanillic acid, no activity was observed in the *M. extorquens*–only cultures. In cultures containing *P. putida*, formaldehyde was generated on vanillic acid during the exponential growth phase, and the duration and peak of the formaldehyde was lower in cultures containing *M. extorquens*. The data in panel (**B**) are from a larger experiment testing *M. extorquens* cultures from different pre-growth conditions and inoculation ratios; full results for that experiment are shown in Appendix A. All data shown here are from cultures initiated in stationary phase, and from *M. extorquens* pre-grown on methanol.

**Figure 5 microorganisms-09-00321-f005:**
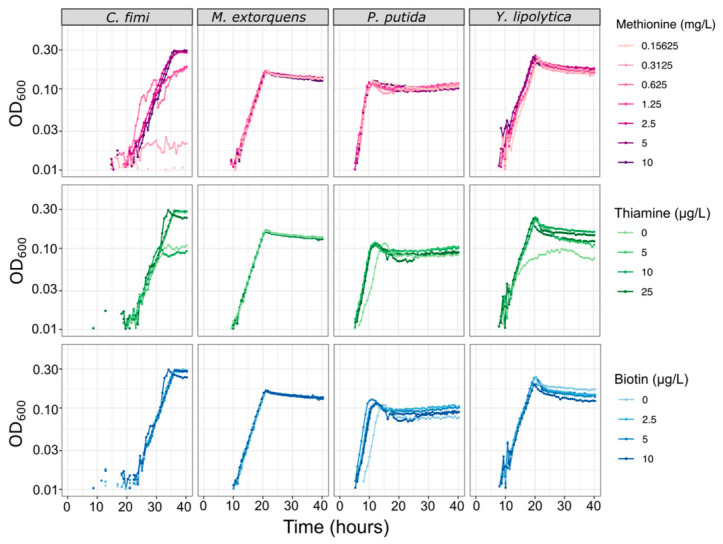
Vitamin and amino acid supplements have minimal effect on growth rates of each of the consortium members, except for *C. fimi.* Each species was grown in pure culture in minimal medium with a range of concentrations of methionine, thiamine, or biotin. For the methionine experiment, all cultures contained 25 µg/L thiamine and 10 µg/L biotin; for the thiamine experiment, 10 mg/L methionine and 10 µg/L biotin; for the biotin experiment, 10 mg/L methionine and 25 µg/L thiamine.

**Figure 6 microorganisms-09-00321-f006:**
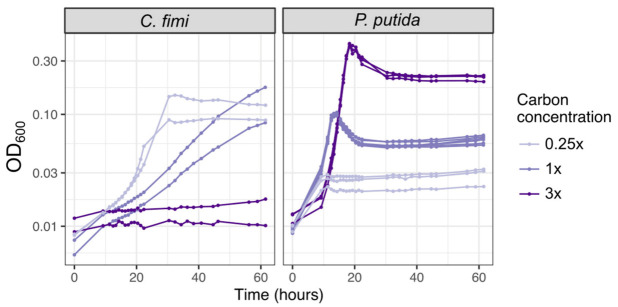
Higher carbon concentrations in model lignocellulose medium result in slower growth and higher final yield; *C. fimi* is more strongly inhibited than *P. putida*. Each species was grown in pure culture in Model Lignocellulose medium (Table 1) with either 1× carbon (4 mM cellobiose, 5 mM xylose, 4 mM vanillic acid), or 0.25 times or 3 times those concentrations.

**Figure 7 microorganisms-09-00321-f007:**
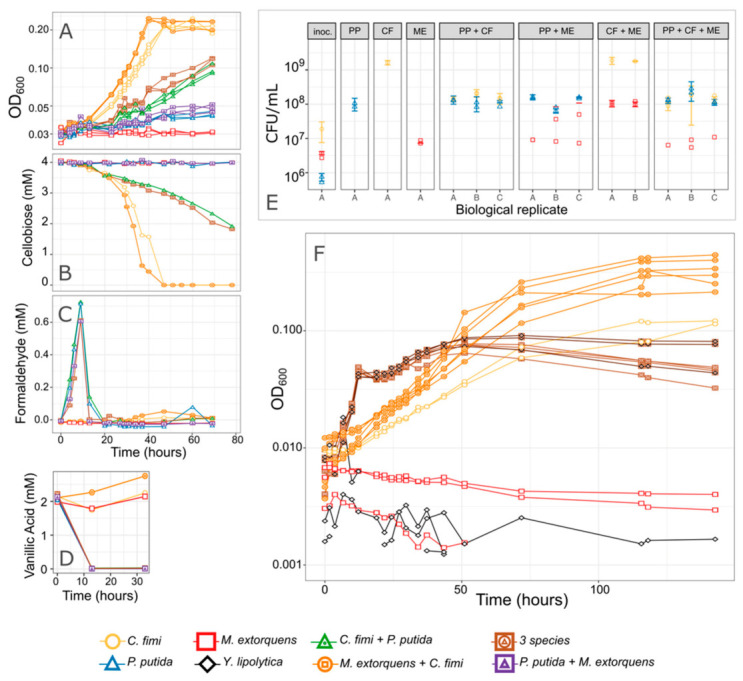
*P. putida* inhibits *C. fimi* growth. (**A**–**E**) Results from an experiment in which each organism was grown in pure culture and in combinations, in minimal medium containing cellobiose and a low concentration of vanillic acid. The combination of species in each culture tube is indicated by the color and shape of the points. (**A**) Growth of the entire community as measured by optical density. Cultures without *P. putida* grew most rapidly and reached the highest final OD. No increase in OD was attributed to *P. putida* because 2 mM vanillic acid supports very little growth. (**B**) Cellobiose concentrations over time. *C. fimi* is the only organism capable of consuming cellobiose, and consumption was most rapid in cultures lacking *P. putida*. (**C**) Formaldehyde concentrations and (**D**) Vanillic acid concentrations over time. Formaldehyde peaked and disappeared, and vanillic acid was consumed, within 12 h, indicating activity by *P. putida* despite fact that the change in OD was not measurable. (**E**) Viable cells from each species, as measured by colony counts, at the beginning of the experiment (“inoc.” = inoculum), and at the end (78 h; each panel is titled with the species present in that culture tube). While no growth substrate was explicitly included to support *M. extorquens*, it increased slightly in abundance by the end of the experiment and the greatest increase was in the presence of *C. fimi* and absence of *P. putida*. Replicate culture tubes are shown along the *x*-axis; points indicate replicate measurements from each tube. (**F**) Results from a similar experiment, testing only a subset of the species combinations in Model Lignocellulose medium. Only optical density is shown here.

**Figure 8 microorganisms-09-00321-f008:**
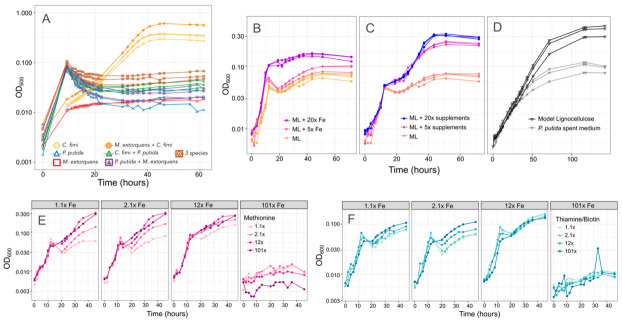
There are multiple potential explanations for the inhibitory effect of *P. putida* on *C. fimi* growth. (**A**) Growth of individual species and combinations (denoted by color and shape of plot symbol) on Model Lignocellulose medium in which vanillic acid is replaced by protocatechuic acid, which does not result in formaldehyde production. Cultures with *P. putida* still show slower growth than those without. (**B**,**C**) Effect of increasing the iron or supplements (methionine, thiamine, and biotin) in the medium. In both cases, higher concentrations support more growth. (**D**) Growth of *C. fimi* alone on either Model Lignocellulose or *P. putida* spent medium: a lower final yield is reached on spent medium. (**E**,**F**) Growth of the model community (*P. putida, C. fimi, M. extorquens*) on combinations of different iron (panels), methionine (color scale), and thiamine + biotin (color scale) concentrations.

**Figure 9 microorganisms-09-00321-f009:**
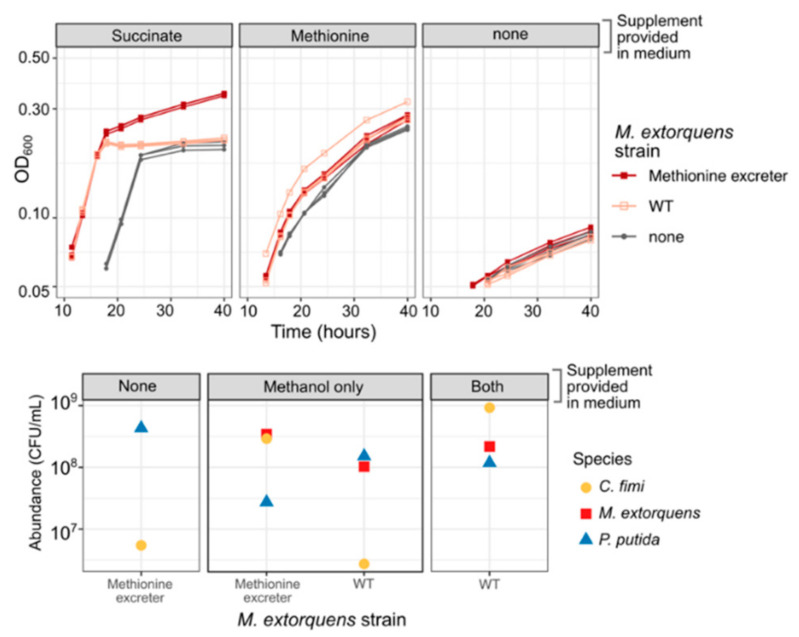
A methionine-overproducing strain of *M. extorquens* can support the growth of *C. fimi* without the addition of methionine to the medium. Top: growth curves of *C. fimi* with different strains of *M. extorquens* (symbols), on MP medium with glucose and supplements (panels). Without succinate, *M. extorquens* growth is negligible. For *C. fimi* alone, growth with methionine is much greater than without. However, when succinate is present to enable *M. extorquens* growth, the culture of *C. fimi* + the methionine excreter reaches substantially higher growth than by *C. fimi* + WT *M. extorquens*, suggesting a benefit to *C. fimi* from the methionine. Bottom: colony counts of each species (symbols) from the endpoint of an experiment in which the three-species consortium was grown on Model Lignocellulose medium with either of the two strains of *M. extorquens* (*x*-axis), and supplemented with either nothing, methanol to support *M. extorquens* growth, or both methanol and methionine (panels). *C. fimi* grows only when methionine is present or when both methanol and the methionine-excreting *M. extorquens* strain are present. *P. putida* does not show the same reliance on methionine or *M. extorquens*.

**Figure 10 microorganisms-09-00321-f010:**
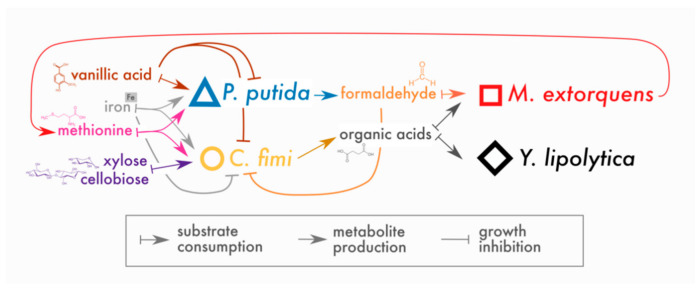
Revised conceptual model of interactions in model lignocellulose-degrading microbial consortium. To our original model, we have added competition between *P. putida* and *C. fimi* for iron and methionine, and inhibitory effects of vanillic acid on *C. fimi* and *P. putida* and high concentrations of iron on *C. fimi*. We have found little evidence for inhibitory effects of formaldehyde (at the concentrations produced here) on any of the members besides *C. fimi*; there is evidence that *C. fimi*, but not *P. putida*, supports *M. extorquens* growth, likely through organic acid production. In addition, we have added the ability for *M. extorquens* to support *C. fimi* through the production of methionine.

**Table 1 microorganisms-09-00321-t001:** Components of Model Lignocellulose medium. Basic MP medium is prepared as described in [26], with the exception that PIPES and P concentrations are altered in Model Lignocellulose: originally published concentrations for these compounds are given in square brackets.

Category	Component	Final Concentration
Basic modified PIPES medium (MP)	PIPES free acid	3 mM [30 mM]
	Potassium phosphate dibasic, K_2_HPO_4_	2.90 mM [1.45 mM]
	Sodium phosphate monobasic, NaH_2_PO_4_	3. 76 mM [1.88 mM]
	Magnesium chloride, MgCl_2_	0.5 mM
	Ammonium sulfate, (NH_4_)_2_SO_4_	5 mM
	Sodium citrate, Na_3_C_6_H_5_O_7_	45.286 µM
	Zinc sulfate, ZnSO_4_	1.2 µM
	Manganese chloride, MnCl_2_	1.02 µM
	Iron(II) sulfate, FeSO_4_	17.768 µM
	Ammonium heptamolybdate, (NH_4_)_6_Mo_7_O_24_	2 µM
	Copper(II) sulfate, CuSO_4_	1 µM
	Cobalt chloride, CoCl_2_	2 µM
	Sodium tungstate, Na_2_WO_4_	0.338 µM
	Calcium chloride, CaCl_2_	0.02 mM
Supplements	Methionine	2 mg/L
	Thiamine	5 µg/L
	Biotin	40 µg/L
Carbon substrates	Cellobiose	4 mM
	Xylose	5 mM
	Vanillic Acid	4 mM

## Data Availability

All data presented in this study are available in Appendix A.

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
