# Peer review of "Cross-Feeding of a Toxic Metabolite in a Synthetic Lignocellulose-Degrading Microbial Community"

_microorganisms, 2021, doi:10.3390/microorganisms9020321_

Round 1

Reviewer 1 Report

This is a rather unique and highly original study, especially as relevant to this special issue of Microorganisms that specifically deals with methylotrophy and its biotechnological applications. The study was well conceived and was performed carefully, producing some expected and some unexpected results. The paper is very well written. Ultimately, the study presents a well developed background for future experiments by the same group or by others. I only have several minor comments toward improving the manuscript.

L 448 bad grammar

L 526 fix ‘as as’

575 perhaps they compete for the same nutrient?

626 This is very speculative, please tone this down

Fig 1; Fig 10 legends need to be expanded to explain your concept/conclusions. For example, what do colors mean? What do the shapes mean? While I understood what you mean by x/y/z, I am pretty sure this could be explained in a much less convoluted way, such as arrow a shows nutrient consumed, arrow b shows compound produced, and arrow c shows inhibition. X/Y/Z are not necessary, especially as the organisms are named.

Fig 6 I would say P. putida grows the same under all conditions. Please comment further if you insist the growth is different

Fig 7 Does this mean that M. extorquens prefers something else, not CH2O?

Reviewer 2 Report

Summary

The manuscript describes the establishment of system for studying lignocellulose degradation using synthetic microbial consortium harboring Pseudomonas putida, Cellulomonas fimi, Methylorubrum exotorquens, and Yarrowia lipolytica. The author first generated a simple defined Model Lignocellulose medium (MLM) to support the growth of all members of synthetic microbial community. Using the MLM, the authors then conducted growth monitoring experiments with vanillic acid and/or formaldehyde. Finally, they provided a conceptual model of interactions in model lignocellulose-degrading microbial consortium.  I feel that this is a preliminary study for efficient lignocellulose degradation using microbes. However, the manuscript provided very useful data to design and build synthetic microbial communities for lignocellulose degradation.

Major Comments

  1. In most Figures of the manuscript is lacking information about the number of independent experiments carried out to obtain the values shown. This information is very important, especially when the differences among samples are not very high. In all these cases, at least three different and independent experiments should be performed and a careful statistical analysis should be included to show whether the observed differences are really significant. In addition, in all figures, the font size of numbers should be larger because it is very hard to read them.

  1. In Fig. 3, formaldehyde tolerance was tested with all four strains. But there is no result of Y. lipolytica in Fig. 3E. 

  1. In Fig. 4, there should be consumption data of vanillic acid along with the formaldehyde production because the authors consistently claimed that the formaldehyde is produced from vanillic acid in P. putida.

  1. In Fig. 7, once the vanillic acid was depleted in approximately 12 hr, where was formaldehyde produced in P. putida? There is no legend for black diamonds in Fig. 7F. It should be Y. lipolytica. The authors should word how to do species specific analysis of cell abundance as CFU in Fig. 7E. 

Round 2

Reviewer 2 Report

Although the authors addressed most of my concerns, the revised manuscript should be improved by providing more results. If the authors want to provide useful information on designing and building synthetic microbial consortium, the authors should carry out the formaldehyde tolerance test of Y. lipolytica that is only a eukaryotic strain (Fig. 3). Except for this, I have no objection to the publication of this revised manuscript.